# GLOBAL MAGNITUDE PRUNING WITH MINIMUM THRESHOLD IS ALL WE NEED

## ABSTRACT

Neural network pruning remains a very important yet challenging problem to solve. Many pruning solutions have been proposed over the years with high degrees of algorithmic complexity. In this work, we shed light on a very simple pruning technique that achieves state-of-the-art (SOTA) performance. We showcase that magnitude based pruning, specifically, global magnitude pruning (GP) is sufficient to achieve SOTA performance on a range of neural network architectures. In certain architectures, the last few layers of a network may get over-pruned. For these cases, we introduce a straightforward method to mitigate this. We preserve a certain fixed number of weights in each layer of the network to ensure no layer is over-pruned. We call this the Minimum Threshold (MT). We find that GP combined with MT when needed, achieves SOTA parameter-accuracy trade-off on all datasets and architectures tested including ResNet-50 and MobileNet-V1 on ImageNet. Code is available at https://github.com/GPMT-Authors/Global-Pruning-With-Minimum-Threshold.

## 1 INTRODUCTION

Neural network pruning remains an important area from both practical perspective (deployment in real world applications) and academic perspective (understanding how to create an efficient architecture). It is a long standing area of exploration (LeCun et al., 1990a; Hassibi & Stork, 1993a), and was reinvigorated by Han et al. (2015a). Since then, much work has been done on trying to find different ways of pruning neural networks, such as magnitude-based, gradient or second order based, and regularization-based methods amongst many others. In this work, we shed light on an often overlooked method that has been seen as a mediocre baseline by the community — global magnitude pruning (GP), and show that it can achieve SOTA pruning performance.

We demonstrate that GP by itself is a strong pruning algorithm and outperforms SOTA pruning algorithms on benchmarks like ResNet-50 and MobileNet-V1 on ImageNet. We also investigate the pruning behavior of GP and find that a simple addition to GP can raise its performance even more. In contrast to the idea of sparsifying each layer of a neural network to the maximum possible level, we find that preserving a certain amount of weights in each layer actually leads to a better pruning scheme, achieving higher accuracy at the same sparsity level. We call this the Minimum Threshold (MT). When combined with GP, this technique enhances the pruning performance in most cases.

We conduct a range of experiments to showcase the above and also study detailed ablations to isolate the effects of GP and MT. We obtain SOTA accuracies on all four sparsity targets on ResNet-50 on ImageNet. We obtain SOTA accuracies on other architectures and datasets tested as well. Finally, GP with MT (GPMT) is very simple conceptually and very easy to implement. It is a one-shot pruning method in which the weights to be pruned are decided in one-go without needing any iterative or gradual phases.

## 2 RELATED WORK

Compression of neural networks has become an important research area due to the rapid increase in size of neural networks (Brown et al., 2020), the need for fast inference on edge devices, e.g., a quadrotor's onboard computer (Camci et al., 2020), and concerns about the carbon footprint of

training large neural networks (Strubell et al., 2019). Over the years, several compression techniques have emerged in the literature (Cheng et al., 2017), such as quantisation, factorisation, attention, knowledge distillation, architecture search and pruning (Almahairi et al., 2016; Ashok et al., 2017; Iandola et al., 2016; Pham et al., 2018b).

Quantisation techniques which restrict the bitwidth of parameters (Rastegari et al., 2016; Courbariaux et al., 2016) and tensor factorisation and decomposition which aim to break large kernels into smaller components (Mathieu et al., 2013; Gong et al., 2014; Lebedev et al., 2014; Masana et al., 2017) are popular methods. However, they need to be optimised for specific architectures. Attention networks (Almahairi et al., 2016) have two separate networks to focus on only a small patch of the input image. Training smaller student networks in a process called knowledge distillation (Ashok et al., 2017) has also proved effective, although it can potentially require a large training budget. Architecture search techniques, such as new kernel design (Iandola et al., 2016) or whole architecture design (Pham et al., 2018a; Tan et al., 2019) have also become popular. Nevertheless, the large search space size requires ample computational resources to do the architecture search. Different from all these approaches, we focus on pruning deep neural networks in this work. As compared to other categories, pruning is more general in nature and has shown strong performance (Gale et al., 2019).

Many pruning techniques have been developed over the years, which use first or second order derivatives (LeCun et al., 1990b; Hassibi & Stork, 1993b), gradient based methods (Lee et al., 2018; Wang et al., 2020), sensitivity to or feedback from some objective function (Molchanov et al., 2017; Liu et al., 2020; Lin et al., 2020; de Jorge et al., 2021), distance or similarity measures (Srinivas & Babu, 2015), regularization-based techniques (Kusupati et al., 2020; Savarese et al., 2020; Wang et al., 2021), and magnitude-based criterion (Ström, 1997; Zhu & Gupta, 2018; Park et al., 2020; Evci et al., 2020; Lee et al., 2021). Han et al. (2015b) discovered a key trick to iteratively prune and retrain the network, thereby preserving high accuracy. Gale et al. (2019) adopt simple, magnitude-based pruning but employ gradual pruning that requires high computational budget and preset sparsification schedules. Runtime Neural Pruning (Lin et al., 2017) attempts to use reinforcement learning (RL) for compression by training an RL agent to select smaller sub-networks during inference. He et al. (2018) design the first approach using RL for pruning. However, RL training approaches typically require additional RL training budgets (Gupta et al. (2020)) or iterative pruning to achieve good accuracy (He et al. (2018)).

In this work, we focus on a simple, effective, yet quite overlooked pruning method — global magnitude pruning (GP). Although first proposed in 1990s (Hoefler et al. (2021)), it has largely been ignored in recent years, generally being relegated to the position of a baseline for comparison (Zhu & Gupta, 2018; Blalock et al., 2020; Lee et al., 2021) rather than a strong pruning technique. A few recent works use it as one in a possible pool of pruning techniques (See et al., 2016; Frankle & Carbin, 2018; Gohil et al., 2020) but never study it in detail or adopt it as the main pruning method. We delve deep into GP and showcase that it can achieve SOTA results with the addition of the MT technique. We present SOTA results on various architectures and datasets including ResNet-50 and MobileNet-V1 on ImageNet, and include comprehensive ablation studies and insights on the workings of GP with MT (GPMT).

An advantage of GPMT is that it is a very simple and reliable approach. There are a few levels of simplicities and robustness to GPMT. Firstly, it is conceptually very simple and easy to implement. It does not require any complex pruning frameworks like RL (He et al., 2018) or sparsification schedules (Zhu & Gupta, 2018). Secondly, it is one-shot and does not require any iterative procedure. Thirdly, it is easily generalizable across architectures and datasets, as shown in the experiments. Lastly, it is data-independent and does not access the dataset for determining the pruning mask.

## 3 METHOD

We conduct unstructured weight pruning using magnitude pruning. Below we describe the key components of our algorithm in more detail.

## 3.1 GLOBAL MAGNITUDE PRUNING (GP)

GP is a magnitude based pruning approach whereby weights bigger than a given threshold are kept and weights smaller than the threshold are pruned.

Formally, for a given threshold $t$ and each individual weight $w$ in any layer, the new weight $w_{new}$ is defined as follows:

$$w_{new} = \begin{cases} 0 & |w| < t, \\ w & otherwise. \end{cases} \tag{1}$$

In contrast to layer-wise pruning, the threshold is not set on a per layer basis but rather a single threshold is set for the entire network. In this aspect, GP is much more efficient than layer-wise pruning because the threshold does not need to be searched for every layer. On the other hand, uniform pruning refers to setting the same sparsity target for each layer. Thus, every layer is pruned by the same percentage.

## 3.2 MINIMUM THRESHOLD (MT)

The Minimum Threshold (MT) refers to a fixed number of weights that are preserved in every layer of the neural network post pruning. The MT is a scalar value that is fixed before the start of the pruning cycle. The weights in a layer are sorted by their magnitude and the top MT number of weights are preserved. For instance, an MT of 500 implies that 500 of the largest weights in every layer need to be preserved post pruning. If a layer is smaller than the MT number, then it implies that all the weights of that layer must be preserved. Therefore, the MT is a very simple concept to apply and also computationally inexpensive. This corresponds to-

$$\min \|W_l\|_0 = \begin{cases} \sigma & \text{if } m \geq \sigma, \\ m & \text{otherwise} \end{cases} \tag{2}$$

where $W_l \in \mathbb{R}^m$ denotes the weight vector for layer $l$, $\sigma$ is the MT value in terms of the number of weights and $\min \|W_l\|_0$ indicates the number of non-zero elements in $W_l$. We explain in the below section how the actual pruning using MT is implemented.

## 3.3 THE PRUNING WORKFLOW

The pruning pipeline for GP and GP with MT (GPMT) is straightforward. It consists of pruning the original model followed by fine-tuning for a few epochs. It is one-shot, therefore, the pruning & fine-tuning cycle does not need to be repeated multiple times. In terms of the pruning procedure itself, for GP, it consists of doing one pass over the network and pruning the weights according to their magnitude to reach the specified sparsity target. For GPMT, it consists of two steps. Firstly, the model is pruned using GP. Secondly, the pruned model is evaluated to check if the MT condition is met by all layers or not. If the condition is not met by a layer, then its sparsity ratio is reduced to meet the MT. The slack arising from the decrease in sparsity is then redistributed amongst the other layers which do not violate the MT condition. The redistribution is done in proportion to their existing sparsities so as to preserve their relative sparsities. This finishes the pruning cycle and the network is then fine-tuned. Fig. 1 explains this procedure, while Algorithm 1 gives the pseudocode.

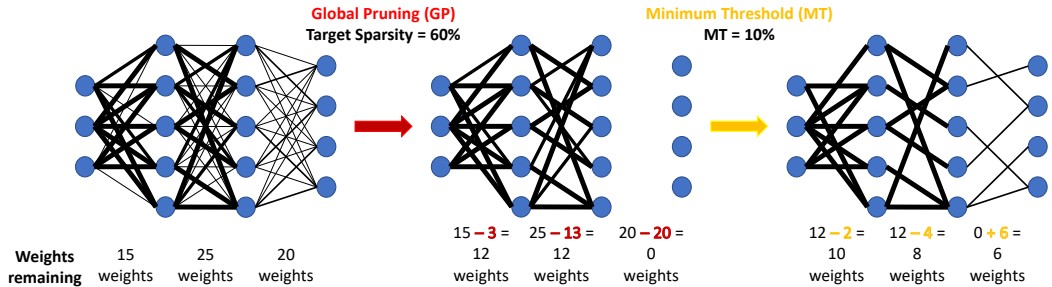

Figure 1: An example workflow over a three-layer network with 60 weights. Thicker lines represent the weights with high magnitude. The network is first pruned to 60% sparsity globally, then MT of 10% (6 weights) is activated per layer. Since the last layer is completely pruned by GP, MT recovers the minimum number of connections there. It distributes the slack arising from the drop in sparsity to the other layers in proportion to their existing sparsities, i.e., 2 weights for Layer 1 (sparsity 20%) and 4 weights for Layer 2 (sparsity 52%). Red color indicates changes in weights due to GP and yellow indicates changes due to MT.

| Method | WRN-22-8 on CIFAR-10 | | |
| --- | --- | --- | --- |
| | Sparsity | Starting Acc. | Pruned Acc. |
| Uniform Pruning | 95% | $94.07\% \pm 0.05\%$ | $94.16\% \pm 0.10\%$ |
| GP | 95% | $94.07\% \pm 0.05\%$ | $94.43\% \pm 0.02\%$ |
| **GP + MT** | 95% | $94.07\% \pm 0.05\%$ | $\mathbf{94.64\% \pm 0.14\%}$ |

Table 1: GP improves performance on Uniform Pruning and adding MT improves performance even further.

| Method | MobileNet-V2 on CIFAR-10 | | |
| --- | --- | --- | --- |
| | Sparsity | Starting Acc. | Pruned Acc. |
| Uniform Pruning | 40% | $94.15\% \pm 0.23\%$ | $93.76\% \pm 0.18\%$ |
| GP | 40% | $94.15\% \pm 0.23\%$ | $94.07\% \pm 0.14\%$ |
| **GP + MT** | 40% | $94.15\% \pm 0.23\%$ | $\mathbf{94.21\% \pm 0.12\%}$ |

Table 2: Same trend is seen for MobileNet as well where GP outperforms Uniform Pruning and adding MT improves performance even further.

## 4 EXPERIMENTS

Below we describe experiments related to ablations on global magnitude pruning (GP) and Minimum Threshold (MT), comparison with state-of-the-art algorithms and experiments on non-vision domains. We report hyper-parameters and training related information for all the experiments in the appendix (section A.4).

### 4.1 ISOLATING AND UNDERSTANDING IMPACT OF GP AND MT OVER DIFFERENT ARCHITECTURES

We conduct detailed ablations to isolate and measure the impact of standalone GP and GP with MT (GPMT) as compared to a uniform pruning baseline. We ablate on multiple architectures and sparsity targets. In addition, we report results averaged over multiple runs where each run uses a different pre-trained model to provide more robustness. We first prune a WRN-22-8 model on CIFAR-10 at 95% sparsity. We then fine-tune the model for a few epochs and report the final accuracy. We experiment with different pruning schemes, i.e., uniform pruning, GP and GPMT (see Section 3 for details). We find that GP outperforms uniform pruning. Furthermore, adding MT improves the performance even more, see Table 1. Next, we do an experiment on the highly efficient MobileNet-V2 architecture to see if the above conclusion holds on it too. We find that indeed GP beats uniform pruning in this situation as well, and adding MT improves performance even further

(Table 2). This shows that GP by itself is superior to uniform pruning and adding MT aids GP even more.

| Method | WRN-22-8 on CIFAR-10 | | |
|--------|---------|---------------|-------------|
| | Sparsity | Starting Acc. | Pruned Acc. |
| GP | 99.9% | $94.07\% \pm 0.05\%$ | $67.68\% \pm 0.78\%$ |
| **GP + MT** | 99.9% | $94.07\% \pm 0.05\%$ | $\mathbf{68.42\% \pm 0.58\%}$ |

Table 3: MT improves performance on WideResNet-22-8 even in the high sparsity regime at 99.9% sparsity.

| Method | MobileNet-V2 on CIFAR-10 | | |
|--------|---------|---------------|-------------|
| | Sparsity | Starting Acc. | Pruned Acc. |
| GP | 98.0% | $94.15\% \pm 0.23\%$ | 10% *(Unable to learn)* |
| GP + MT | 98.0% | $94.15\% \pm 0.23\%$ | $82.97\% \pm 0.57\%$ |
| **Gradual GP** | **98.0%** | NA | $\mathbf{87.36\% \pm 0.18\%}$ |

Table 4: Adding MT or conducting GP gradually enables the MobileNet model to learn and achieve good classification performance in the high sparsity regime.

We then conduct experiments under much tougher conditions, increasing the sparsity rates on both the above-mentioned models to see how the algorithm performs. We find that at 99.9% sparsity, the WRN is able to get decent accuracy and adding MT improves performance even at this sparsity rate (Table 3). For MobileNet however, using GP only, accuracy drops to 10% and the model is not able to learn. We discuss this in detail in Section 5.1. However, on adding the MT, the model is able to learn and the accuracy jumps back to 83% (Table 4). Alternatively, conducting GP gradually (Zhu & Gupta, 2018) also enables the model to learn well and achieves 87.36% accuracy. We provide layer-wise weight snapshot for both the models before and after applying MT, to illustrate what MT does (Fig. 2 and Fig. 3). Thus, in cases of high sparsity, MT is desirable to have.

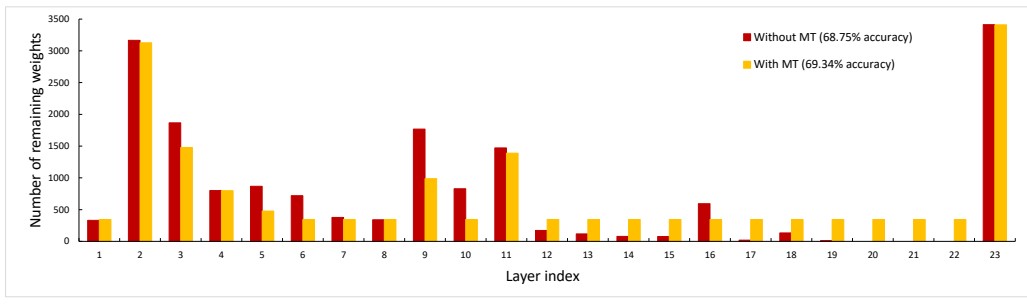

Figure 2: Weights remaining in WRN-22-8 model after pruning at 99.9% sparsity. MT helps retain weights in layers that have low magnitude weights and are heavily pruned, e.g., Layers 20, 21, and 22.

## 4.2 STATE-OF-THE-ART COMPARISON ON CIFAR-10

We also conduct experiments to compare GP and MT to SOTA pruning algorithms on the CIFAR-10 dataset. We compare with various algorithms including SNIP (Lee et al., 2018), SM (Dettmers & Zettlemoyer, 2019), DSR (Mostafa & Wang, 2019) and DPF (Lin et al., 2020). We start off with the original model having the same initial accuracy as the other algorithms. For the WRN-28-8 model we find that GP alone beats all the SOTA pruning algorithms at two out of the three sparsity levels (Table 5). If we incorporate MT, then the accuracy increases further, and the algorithm is able to beat all the SOTA algorithms at all the sparsity levels. For ResNet-32 as well (Table 6), both GP and GPMT outperform all the other algorithms at 90% sparsity.

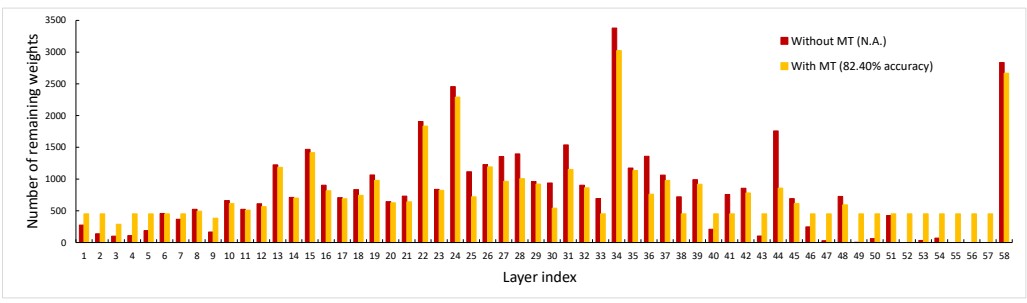

Figure 3: For MobileNet-V2 at 98% sparsity as well, MT is essential to retain some weights in the heavily pruned layers (Layers 55, 56, and 57) and allow the model to learn successfully.

### 4.3 STATE-OF-THE-ART COMPARISON ON IMAGENET

We apply GP and MT on ResNet-50 on the ImageNet dataset as well and compare to the SOTA pruning algorithms. We compare with SOTA algorithms like GMP (Zhu & Gupta, 2018), DSR (Mostafa & Wang, 2019), DNW (Wortsman et al., 2019), SM (Dettmers & Zettlemoyer, 2019), RigL (Evci et al., 2020), DPF (Lin et al., 2020) and STR (Kusupati et al., 2020). We start with the same original accuracy as the other models. We find that GP alone gets good performance and surpasses all the other algorithms at all the sparsity levels (see Table 7) keeping the numbers of parameters constant. Adding MT leads to roughly the same performance as GP (slightly higher and slightly lower in certain cases) and this is explained by the fact that all layers in ResNet-50 always maintain enough weights (i.e., $\geq 1,000$) at all sparsity levels. GP incurs higher FLOPs compared to some baselines as it prunes the last layers more vs. the initial layers (which have a higher FLOPs to parameters ratio) compared to some baselines. Optimization for parameters vs. FLOPs is usually a trade-off against each other and joint optimization of parameters and FLOPs can be implemented as future work to reduce the FLOPs of GP. GP still outperforms other baselines like DSR, SM, SM + ERK and RigL + ERK on both accuracy and FLOPs.

We also test another architecture on ImageNet, MobileNet-V1, which is a much smaller and more efficient architecture than ResNet-50. We start with the same original accuracy as the other models. For MobileNet-V1, GP alone gets good results at 75% sparsity and outperforms SOTA algorithms by 2.44% on a constant parameter budget basis (Table 8). On combining with MT, GPMT also surpasses the SOTA algorithms at 90% sparsity by 2.43%. Since MobileNet-V1 is a very efficient architecture (having only 4.2 million parameters compared to 25.6 million for ResNet-50), MT especially helps at higher sparsities when the layers are pruned very aggressively.

### 4.4 GENERALIZING TO OTHER DOMAINS AND RNN ARCHITECTURES

We experiment with the GP rule on other domains and non-convolutional networks as well to measure the generalizability of the algorithm on different domains and network types. We experiment on a FastGRNN model (Kusupati et al. (2018)) on the HAR-2 Human Activity Recognition dataset (Anguita et al. (2013)). More details on the dataset and the experimental setup can be found in Section A.3. We test the GP rule under different network configurations. We find that GP surpasses other pruning algorithms on all the configurations (Table 9) and successfully prunes the model on a very different architecture and domain.

## 5 INSIGHTS ON GP AND MT

### 5.1 NETWORK ARCHITECTURES CAN AFFECT PRUNING PERFORMANCE

We find that network architectures can affect the results of how the pruning algorithm performs quite a lot, especially in the high sparsity domain. We take the case of the high sparsity experiments

between MobileNet-V2 and WRN-22-8, as mentioned in Section 4.1. In the case of just using GP, we find that WRN-22-8 is still able to learn, however, accuracy crashes for MobileNet-V2 and the model is unable to learn anything, getting a chance accuracy rate of 10%. The reason for this wide discrepancy in learning behavior lies in the shortcut connections (He et al., 2015). Both WRN-22-8 and MobileNet-V2 use shortcut connections, however, their placement is different. Referring to Fig. 4, WRN uses identity shortcut connections from Layer 20 to Layer 23. Identity shortcut connections are simple identity mappings and do not require any extra parameters, and hence, they do not count towards the weights. However, MobileNet-V2 uses a convolutional shortcut mapping from Layer 52 to Layer 57 and hence, it does add to the model's weights, and thus, it is pruned by the pruning algorithm. Both the models have the preceding two layers before the last layer, completely pruned. However, because WRN uses identity mappings, it is still able to relay information to the last layer, and the model is still able to learn.

| Method | Top-1 Acc | Params. | Sparsity |
|---|---|---|---|
| WRN-28-8 | 96.06% | 23.3M | 0.0% |
| SNIP | $95.49 \pm 0.21\%$ | 2.33M | 90% |
| SM | $95.67 \pm 0.14\%$ | 2.33M | 90% |
| DSR | $95.81 \pm 0.10\%$ | 2.33M | 90% |
| DPF | $96.08 \pm 0.15\%$ | 2.33M | 90% |
| **GP** | $\mathbf{96.30 \pm 0.03\%}$ | 2.33M | 90% |
| **GP + MT** | $\mathbf{96.44 \pm 0.09\%}$ | 2.33M | 90% |
| SNIP | $94.93 \pm 0.13\%$ | 1.17M | 95% |
| SM | $95.64 \pm 0.07\%$ | 1.17M | 95% |
| DSR | $95.55 \pm 0.12\%$ | 1.17M | 95% |
| DPF | $95.98 \pm 0.10\%$ | 1.17M | 95% |
| **GP** | $\mathbf{96.16 \pm 0.02\%}$ | 1.17M | 95% |
| **GP + MT** | $\mathbf{96.27 \pm 0.06\%}$ | 1.17M | 95% |
| SNIP | $94.11 \pm 0.19\%$ | 0.58M | 97.5% |
| SM | $95.31 \pm 0.20\%$ | 0.58M | 97.5% |
| DSR | $95.11 \pm 0.07\%$ | 0.58M | 97.5% |
| DPF | $95.84 \pm 0.04\%$ | 0.58M | 97.5% |
| GP | $95.68 \pm 0.08\%$ | 0.58M | 97.5% |
| **GP + MT** | $\mathbf{95.89 \pm 0.02\%}$ | 0.58M | 97.5% |

Table 5: Results of SOTA pruning algorithms on WideResNet-28-8 on CIFAR-10. GP + MT (GPMT) outperforms all the other algorithms.

| Method | Top-1 Acc | Params. | Sparsity |
|---|---|---|---|
| ResNet-32 | $93.83 \pm 0.12\%$ | 0.46M | 0.00% |
| SNIP | $90.40 \pm 0.26\%$ | 0.046M | 90% |
| SM | $91.54 \pm 0.18\%$ | 0.046M | 90% |
| DSR | $91.41 \pm 0.23\%$ | 0.046M | 90% |
| DPF | $92.42 \pm 0.18\%$ | 0.046M | 90% |
| **GP** | $\mathbf{92.67 \pm 0.03\%}$ | 0.046M | 90% |
| **GP + MT** | $\mathbf{92.74 \pm 0.06\%}$ | 0.046M | 90% |
| SNIP | $87.23 \pm 0.29\%$ | 0.023M | 95% |
| SM | $88.68 \pm 0.22\%$ | 0.023M | 95% |
| DSR | $84.12 \pm 0.32\%$ | 0.023M | 95% |
| **DPF** | $\mathbf{90.94 \pm 0.35\%}$ | 0.023M | 95% |
| GP | $90.65 \pm 0.13\%$ | 0.023M | 95% |
| GP + MT | $90.58 \pm 0.24\%$ | 0.023M | 95% |

Table 6: Results of pruning algorithms on ResNet-32 on CIFAR-10. GPMT outperforms all the other algorithms at 90% sparsity.

| Method | Top-1 Acc | Params. | Sparsity | FLOPs |
|---|---|---|---|---|
| MobileNet-V1 | 71.95% | 4.21M | 0.00% | 569M |
| GMP | 67.70% | 1.09M | 74.11% | 163M |
| STR | 68.35% | 1.04M | 75.28% | 101M |
| **GP** | **70.74%** | 1.04M | 75.28% | 177M |
| **GP + MT** | **70.79%** | 1.04M | 75.28% | 204M |
| GMP | 61.80% | 0.46M | 89.03% | 82M |
| STR | 61.51% | 0.44M | 89.62% | 40M |
| GP | 59.49% | 0.42M | 90.00% | 93M |
| **GP + MT** | **63.94%** | 0.42M | 90.00% | 154M |

Table 8: Results of pruning algorithms on MobileNet-V1 on ImageNet. GPMT surpasses the SOTA algorithms by 2.4% accuracy.

| Method | Top-1 Acc | $r_W$ | $r_U$ |
|---|---|---|---|
| FastGRNN | 96.10% | 9 | 80 |
| Vanilla Training | 94.06% | 9 | 8 |
| STR | 95.76% | 9 | 8 |
| **GP** | **95.89%** | 9 | 8 |
| Vanilla Training | 93.15% | 9 | 7 |
| STR | 95.62% | 9 | 7 |
| **GP** | **95.72%** | 9 | 7 |
| Vanilla Training | 94.88% | 8 | 7 |
| STR | 95.59% | 8 | 7 |
| **GP** | **95.62%** | 8 | 7 |

Table 9: Results of pruning algorithms on FastGRNN on HAR-2 dataset. GP outperforms other pruning algorithms in all the different network configurations.

| Method | Top-1 Acc | Params. | Sparsity | FLOPs | Method | Top-1 Acc | Params. | Sparsity | FLOPs |
|---|---|---|---|---|---|---|---|---|---|
| ResNet-50 | 77.0% | 25.6M | 0.00% | 4.09G | ResNet-50 | 77.0% | 25.6M | 0.00% | 4.09G |
| GMP | 75.60% | 5.12M | 80.00% | 818M | GMP | 70.59% | 1.28M | 95.00% | 204M |
| DSR*# | 71.60% | 5.12M | 80.00% | 1.23G | DNW | 68.30% | 1.28M | 95.00% | 204M |
| DNW | 76.00% | 5.12M | 80.00% | 818M | RigL* | 67.50% | 1.28M | 95.00% | 317M |
| SM | 74.90% | 5.12M | 80.00% | - | RigL + ERK | 70.00% | 1.28M | 95.00% | 600M |
| SM + ERK | 75.20% | 5.12M | 80.00% | 1.68G | STR | 70.97% | 1.33M | 94.80% | 182M |
| RigL* | 74.60% | 5.12M | 80.00% | 920M | STR | 70.40% | 1.27M | 95.03% | 159M |
| RigL + ERK | 75.10% | 5.12M | 80.00% | 1.68G | STR | 70.23% | 1.24M | 95.15% | 162M |
| DPF | 75.13% | 5.12M | 80.00% | 818M | **GP** | **71.56%** | 1.20M | 95.30% | 437M |
| STR | 76.19% | 5.22M | 79.55% | 766M | **GP + MT** | **71.57%** | 1.20M | 95.30% | 438M |
| **GP** | **76.84%** | 5.12M | 80.00% | 1.13G | GMP | 57.90% | 0.51M | 98.00% | 82M |
| **GP + MT** | **76.75%** | 5.12M | 80.00% | 1.28G | DNW | 58.20% | 0.51M | 98.00% | 82M |
| GMP | 73.91% | 2.56M | 90.00% | 409M | STR | 61.46% | 0.50M | 98.05% | 73M |
| DNW | 74.00% | 2.56M | 90.00% | 409M | **GP** | **61.80%** | 0.50M | 98.05% | 257M |
| SM | 72.90% | 2.56M | 90.00% | 1.63G | **GP + MT** | **61.90%** | 0.50M | 98.05% | 257M |
| SM + ERK | 72.90% | 2.56M | 90.00% | 960M | | | | | |
| RigL* | 72.00% | 2.56M | 90.00% | 515M | | | | | |
| RigL + ERK | 73.00% | 2.56M | 90.00% | 960M | | | | | |
| DPF# | 74.55% | 4.45M | 82.60% | 411M | | | | | |
| STR | 74.73% | 3.14M | 87.70% | 402M | | | | | |
| **GP** | **75.28%** | 2.56M | 90.00% | 704M | | | | | |
| **GP + MT** | **75.21%** | 2.56M | 90.00% | 926M | | | | | |

Table 7: Results on ResNet-50 on ImageNet. GP and GPMT outperform SOTA pruning algorithms by upto 1.3% accuracy. * and # imply that the first and last layer are dense respectively.

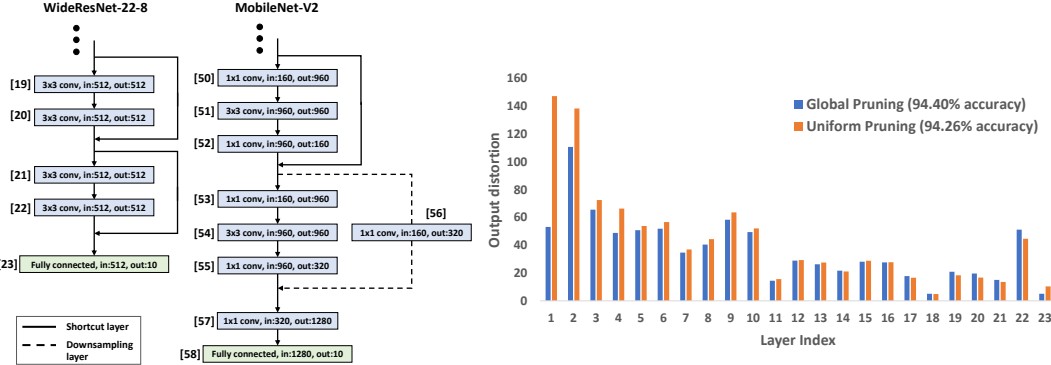

Figure 4: Difference in architectures between WRN and MobileNet. WRN does not have any prunable residual connections in the last layers (dotted lines) while MobileNet does. This leads to different pruning behaviors on the two architectures.

Figure 5: The frobenius output distortion is lower for GP compared to uniform pruning on a layer by layer basis. Thus, GP preserves outputs closer to the original unpruned model compared to uniform pruning. Results on WRN-22-8 on CIFAR-10 dataset.

Pruning algorithms can be susceptible to such catastrophic network disconnection issues especially in the high sparsity domain. Fortunately, the MT can easily overcome this issue. Retaining a small MT of 0.02% was sufficient for the MobileNet-V2 model to avoid disconnection and learn successfully. Hence, retaining a small MT can help in the learning dynamics of models in high sparsity settings.

## 5.2 GP PRESERVES OUTPUTS BETTER THAN UNIFORM PRUNING

To understand why GP performs better than uniform pruning, especially at the layer-by-layer level, we measure the Frobenius output distortion between the original model and the pruned model (Park

et al., 2020). We compare the difference in the layer-wise outputs between the original model and the pruned model, and then compute the norm of this difference. Thus, a smaller norm indicates that the output of the pruned model is closer to the original model and a larger norm indicates that the output is further away. We compute and plot this layer-wise output distortion on the WRN-22-8 model on CIFAR-10 for the GP and uniform pruning models as compared to the original model (Fig. 5). As can be seen from the figure, the distortion is much higher for the uniform pruning model. Layers 1 and 2 are especially impacted, with the distortion being around 3x that of GP in layer 1. Thus, GP is able to preserve outputs much closer to the original model as compared to uniform pruning. This points towards a new direction for empirically understanding the underlying differences between pruning algorithms.

### 5.3 WHEN IS MT REQUIRED AND HOW TO DETERMINE THE OPTIMAL VALUE

We find that whether MT is required or not depends on the neural network architecture family. For instance, in our experiments on multiple architectures, we found generally that ResNet architectures did not require MT (tested on ResNet-32 and ResNet-50). Conversely, MobileNet and WideRes-Net architectures did require MT (tested on MobileNet-V1, MobileNet-V2, WideResNet-22-8 and WideResNet-28-8). Hence, we hypothesize that the requirement for MT depends upon the base family architecture used. We also found that in architectures that require MT, a decent rule of thumb value for MT is 0.05% of the total weights. Increments of 0.05% may be tried, i.e., 0.1%, 0.15% or 0.2%, if need be. We never had to go beyond 0.2%. In cases of high sparsity, where there is not enough capacity in the network to support 0.05% MT, lower values of MT can be tried, e.g., 0.02%, 0.01%, 0.005% or 0.002%. We found that the above-stated values were sufficient to get good performance across models and we never had to undertake a more exhaustive MT search. In most cases, we only tried two to three MT values and achieved good results.

## 6 LIMITATIONS AND FUTURE WORK

A limitation of the GP method is that it can incur higher FLOPs compared to some layer-wise baselines (Table 7). This is because it prunes the last layers more with respect to the initial layers (that have a higher FLOPs to parameter ratio) compared to some baselines. In applications where FLOPs reduction is equally important as parameter reduction, a way to mitigate this would be to add constraints on the FLOPs as well along with the parameters, and thus jointly optimize for both FLOPs and parameters. We will look at this in future work. Another exciting extension to GP is to do GP gradually instead of one-shot. As demonstrated in Table 4, doing GP gradually enhances the accuracy performance of GP. We believe this increase in performance is transferable to other datasets and architectures as well, and is another topic that we will look at in the future.

## 7 CONCLUSION

Many methods have been proposed for neural network pruning over the years. In this paper, we focus on an often overlooked pruning method — global magnitude pruning (GP). We show that GP by itself is a strong pruning method and achieves SOTA performance on ResNet-50 and MobileNet-V1 on ImageNet. Furthermore, we investigate the pruning behavior of GP in depth and find that in certain cases, e.g., at high sparsities, a few layers in the network may be over-pruned. To rectify this, we propose a novel yet very simple mitigation mechanism called Minimum Threshold (MT). MT preserves a certain fixed amount of weights in every layer post pruning and ensures no layer is over-pruned. We find that adding MT on top of GP leads to higher performance across many architectures and datasets. We achieve SOTA results on CIFAR-10 and ImageNet using the above pruning technique. GP with MT (GPMT) is also easy to implement, is one-shot and is easily portable across architectures and datasets. GP can be further extended to be done gradually instead of one-shot and can lead to more performance gains. A limitation of GPMT is that it can incur higher FLOPs compared to some layerwise methods and a possible mitigation for this is to add a FLOPs constraint on top of the parameter constraint when doing pruning. Overall, we find that GPMT can be used as a useful baseline for future pruning works given it's simplicity and performance.

## 8    REPRODUCIBILITY

To help with reproducibility, we have released our codebase. It can be found at https://github.com/GPMT-Authors/Global-Pruning-With-Minimum-Threshold. Furthermore, we report in detail the experimental setup including all the hyper-parameters for all our experiments in the appendix (Section A.4). The above two items ensure that all our experiments are reproducible. Furthermore, our method is conceptually very simple and hence, very easy to implement. It does not require additional algorithm-specific hyper-parameters or any architecture-specific tweaking. We use standard PyTorch libraries (Paszke et al. (2019)) without any modification to the model layers. These make the GPMT method straightforward to implement, and hence, easy to replicate. Lastly, we show that GPMT gets stable results across multiple runs and pre-trained models (Figs. 6 and 7) and hence, aids in reproducibility.

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

# A APPENDIX

## A.1 SIMPLE ONE-SHOT APPROACH

## A.2 PSEUDO-CODE

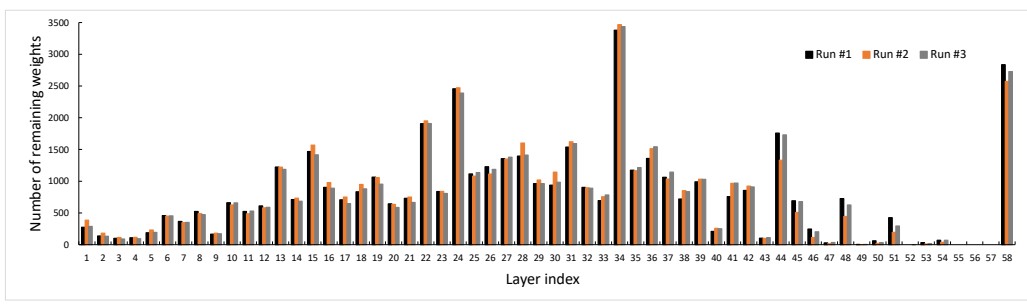

Figure 6: Layer-wise pruning results produced by GP on MobileNet-V2 model on CIFAR-10. Pruning is done on three different pre-trained models and the pruning results across the three runs are very stable.

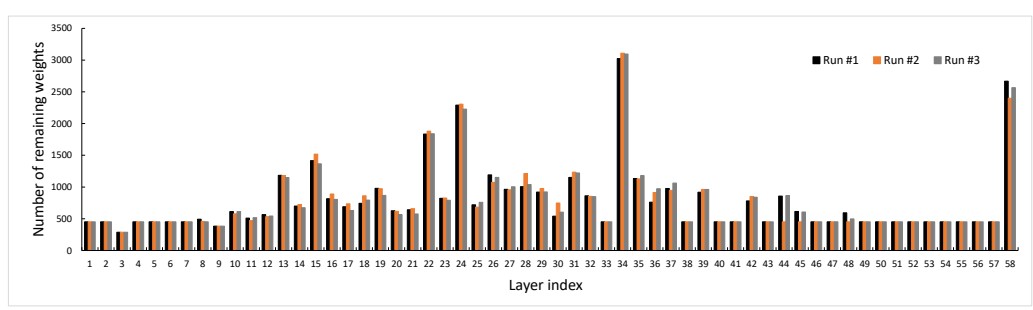

Figure 7: A similar trend is observed for the case of GPMT on MobileNet-V2 model on CIFAR-10 as well. Pruning results on the three different pre-trained models are very stable.

---

**Algorithm 1** Pseudo-code for GPMT

---

Load a pre-trained DNN with weight vectors $W_l$ for each layer $l$
Set a target sparsity, $\kappa$
Set an MT value, $\sigma$
Concatenate and sort all $W_l$ into a vector $\mathbf{W} = \{w_0, w_1, ..., w_n\}$ where $w_i \leq w_{i+1}$
$k = \frac{\kappa}{100} \cdot n$
Zero-mask all elements from $w_0$ to $w_k$ in $\mathbf{W}$ across all $W_l$
Slack pruning budget, $\Sigma = 0$
**for** $l$ **in** layers **do**
   **if** $\min \|W_l\|_0 < \sigma$ **then**
      Restore the $(\sigma - \min \|W_l\|_0)$ # of highest magnitude weights in $W_l$
      $\Sigma = \Sigma + \sigma - \min \|W_l\|_0$
   **end if**
**end for**
Distribute $\Sigma$ into each layer where $\min \|W_l\|_0 > \sigma$, proportionally to their sparsities $\kappa_l$
Fine-tune the pruned model

---

### A.3 SPARSIFYING FASTGRNN ON HAR-2 DATASET

HAR-2 dataset that is used in the FastGRNN pruning experiment is a binarized version of the 6-class Human Activity Recognition dataset. From the full-rank model with $r_W = 9$ and $r_U = 80$ as suggested on the STR paper (Kusupati et al., 2020), we apply GP on the matrices $W_1$ and $W_2$. To do this, we find the weight mask by ranking the columns of $W_1$ and $W_2$ based on their absolute sum, then we prune the $9 - r_W^{new}$ lowest columns and $80 - r_U^{new}$ lowest columns from $W_1$ and $W_2$

respectively. In the end, we fine-tuned this pruned model by retraining it with FastGRNN's trainer and applying the weight mask on every epoch.

## A.4 HYPER-PARAMETERS AND EXPERIMENTAL SETUP

No data augmentation is done apart from standard data pre-processing. Difference in batch size for training and testing in some experiments is due to GPU RAM availability. Averages reported over three runs.

| MT values | | |
| --- | --- | --- |
| Experiment | Sparsity | MT value |
| Table 1 | 95% | 0.05% |
| Table 2 | 40% | 0.05% |
| Table 3 | 99.9% | 0.002% |
| Table 4 | 98% | 0.02% |
| Table 5 | 90% | 0.05% |
| | 95% | 0.10% |
| | 97.5% | 0.05% |
| Table 6 | 90% | 0.05% |
| | 95% | 0.05% |
| Table 7 | 80% | 0.05% |
| | 90% | 0.05% |
| | 95.3% | 0.005% |
| | 98.05% | 0.005% |
| Table 8 | 75% | 0.05% |
| | 90% | 0.2% |

| Setup for Table 1 | | | | | | |
| --- | --- | --- | --- | --- | --- | --- |
| Stage | Epochs | Batch-size | Mom-entum | Weight Decay | Initial LR | LR Scheduler | Nes-terov |
| Training | 30 | 256 | 0.9 | 5e-4 | 0.1 | Step decay (Step size 25, gamma 0.1) | Yes |
| Finetuning | 80 | 1800 | 0.9 | 5e-4 | 0.1 | Step decay (Step size 40, gamma 0.1) | Yes |

| Setup for Table 2 | | | | | | |
| --- | --- | --- | --- | --- | --- | --- |
| Stage | Epochs | Batch-size | Mom-entum | Weight Decay | Initial LR | LR Scheduler | Nes-terov |
| Training | 200 | 450 | 0.9 | 5e-4 | 0.1 | Step decay (Step size 25, gamma 0.56) | Yes |
| Finetuning | 200 | 325 | 0.9 | 5e-4 | 0.031 | Step decay (Step size 25, gamma 0.56) | Yes |

| Setup for Table 3 | | | | | | | |
|---|---|---|---|---|---|---|---|
| Stage | Epochs | Batch-size | Mom-entum | Weight Decay | Initial LR | LR Scheduler | Nes-terov |
| Training | 30 | 256 | 0.9 | 5e-4 | 0.1 | Step decay (Step size 25, gamma 0.1) | Yes |
| Finetuning | 80 | 64 | 0.9 | 5e-4 | 0.1 | Step decay (Step size 40, gamma 0.1) | Yes |

| Setup for Table 4 | | | | | | | |
|---|---|---|---|---|---|---|---|
| Stage | Epochs | Batch-size | Mom-entum | Weight Decay | Initial LR | LR Scheduler | Nes-terov |
| Training | 200 | 450 | 0.9 | 5e-4 | 0.1 | Step decay (Step size 25, gamma 0.56) | Yes |
| Finetuning | 200 | 64 | 0.9 | 5e-4 | 0.1 | Step decay (Step size 25, gamma 0.56) | Yes |

| Setup for Table 5 | | | | | | | |
|---|---|---|---|---|---|---|---|
| Stage | Epochs | Batch-size | Mom-entum | Weight Decay | Initial LR | LR Scheduler | Nes-terov |
| Training | 200 | 128 | 0.875 | 5e-4 | 0.1 | Cosine LR | Yes |
| Finetuning (GP 90%) | 200 | 128 | 0.9 | 0 | 0.0512 | Cosine LR | Yes |
| Finetuning (GP + MT 90%) | 200 | 128 | 0.9 | 5e-4 | 0.0064 | Cosine LR | Yes |
| Finetuning (GP 95%) | 200 | 128 | 0.9 | 2e-5 | 0.0256 | Cosine LR | Yes |
| Finetuning (GP + MT 95%) | 200 | 128 | 0.9 | 5e-4 | 0.0064 | Cosine LR | Yes |
| Finetuning (GP 97.5%) | 200 | 128 | 0.9 | 0 | 0.0128 | Cosine LR | Yes |
| Finetuning (GP + MT 97.5%) | 200 | 128 | 0.9 | 6e-5 | 0.0256 | Cosine LR | Yes |

| Setup for Table 6 | | | | | | | |
|---|---|---|---|---|---|---|---|
| Stage | Epochs | Batch-size | Mom-entum | Weight Decay | Initial LR | LR Scheduler | Nes-terov |
| Training | 300 | 128 | 0.9 | 0.001 | 0.05 | Cosine LR | No |
| Finetuning (GP 90%) | 300 | 128 | 0.9 | 0.001 | 0.01 | Cosine LR | No |
| Finetuning (GP + MT 90%) | 300 | 128 | 0.9 | 0.001 | 0.01 | Cosine LR | Yes |
| Finetuning (GP 95%) | 300 | 128 | 0.9 | 1e-5 | 0.01 | Cosine LR | No |
| Finetuning (GP + MT 95%) | 300 | 128 | 0.875 | 0.0005 | 0.005 | Linear LR | No |

| Setup for Table 7 | | | | | | | |
|---|---|---|---|---|---|---|---|
| Stage | Epochs | Batch-size | Mom-entum | Weight Decay | Initial LR | LR Scheduler | Label Smooth-ing |
| Training | 100 | 256 | 0.875 | 0.000031 | 0.256 | Cosine LR (warmup=5) | 0.1 |
| Finetuning (GP 80%) | 100 | 256 | 0.875 | 0.000023 | 0.0256 | Cosine LR (warmup=5) | 0.1 |
| Finetuning (GP + MT 80%) | 100 | 256 | 0.875 | 0.000023 | 0.0256 | Cosine LR (warmup=5) | 0.1 |
| Finetuning (GP 90%) | 100 | 256 | 0.875 | 0.000007 | 0.1024 | Cosine LR | 0.1 |
| Finetuning (GP + MT 90%) | 100 | 256 | 0.875 | 0.000007 | 0.0512 | Cosine LR | 0.1 |
| Finetuning (GP 95.3%) | 100 | 256 | 0.95 | 0.0 | 0.0512 | Cosine LR | 0.05 |
| Finetuning (GP + MT 95.3%) | 100 | 256 | 0.95 | 0.0 | 0.0512 | Cosine LR | 0.05 |
| Finetuning (GP 98.05%) | 100 | 256 | 0.95 | 0.0 | 0.0512 | Cosine LR | 0.05 |
| Finetuning (GP + MT 98.05%) | 100 | 256 | 0.95 | 0.0 | 0.0512 | Cosine LR | 0.05 |

| Setup for Table 8 | | | | | | | |
|---|---|---|---|---|---|---|---|
| Stage | Epochs | Batch-size | Mom-entum | Weight Decay | Initial LR | LR Scheduler | Label Smooth-ing |
| Training | 100 | 256 | 0.875 | 3.1e-5 | 0.256 | Cosine LR (warmup=5) | 0.1 |
| Finetuning (GP 75%) | 120 | 256 | 0.875 | 1e-5 | 0.0512 | Cosine LR | 0.1 |
| Finetuning (GP + MT 75%) | 120 | 256 | 0.875 | 1e-5 | 0.0512 | Cosine LR | 0.1 |
| Finetuning (GP 90%) | 120 | 256 | 0.875 | 1e-5 | 0.0256 | Cosine LR | 0.1 |
| Finetuning (GP + MT 90%) | 120 | 256 | 0.875 | 0 | 0.0512 | Cosine LR | 0.1 |

| Setup for Table 9 | | | | | |
|---|---|---|---|---|---|
| Stage | Epochs | Batch-size | Initial LR | hd | Optimizer |
| Training | 300 | 100 | 0.0064 | 80 | Adam |
| Finetuning (9,8) | 300 | 64 | 0.5 | 80 | Adam |
| Finetuning (9,7) | 300 | 100 | 0.5 | 80 | Adam |
| Finetuning (8,7) | 300 | 100 | 0.55 | 80 | Adam |

