# OpenReview forum: "Global Magnitude Pruning With Minimum Threshold Is All We Need"
_ICLR.cc/2022/Conference — ICLR 2022 Submitted_

### Official Review · Reviewer_MHY2 · 2021-10-16

**Correctness:** 3
**Technical Novelty And Significance:** 2
**Empirical Novelty And Significance:** 2
**Recommendation:** 3
**Confidence:** 5

**Main Review:**

I will go sequentially in the paper but will start with the strengths follow it up with the weaknesses:

Overall, the writing quality is OK and is easy to follow. The information has been well put together for a new reader to understand without having to worry about the background. The authors should be appreciated for that.

Strengths:
1) The fact that authors revisited GP to show its power is a very nice step. GP has been (according to me) the strongest baselines for pruning for a given parameter budget even now (has been acknowledged in multiple papers).
2) The idea of MT makes sense and is a valid fix for the overly pruned layers in general.
3) The experiments on ImageNet are commended because the insights from there a more valuable than other things presented in this paper.
4) The discussion on network architectures, outputs and settings of MT are interesting in general.
5) I really liked the related work, it is thorough from a literature review standpoint and would recommend new readers to go through it for a nice picture of the field

Weaknesses:
This might sound a bit critical, but I would appreciate it if the authors understand the merit of whatever I am trying to put forward here.

1) No one ever claimed GP is not sufficient to achieve SOTA performance on pruning. GP is extremely powerful no matter what gives the highest accuracy (often) for the same #params even at the highest sparsity levels. GP is not forgotten or overlooked by any means. The reason why people work on layer-wise pruning is slightly different and I will come to it later.
2) While the authors mentioned people of LTH community probably used GP as one of the potential pruning techniques, GP was the only one that resulted in strong tickets at higher sparsities, making IMP explicitly work on GP no matter what other factors are.
3) MT comes into the picture only when there is one-shot pruning and GP has been shown to work great even with gradual schedules (IMP is an example) and will not suffer from pruning out entire layers just because of the gradual processing. I would even argue setting the small pruning schedule is data-independent and has a similar cost as finding the optimal MT as done in this paper. MT makes sense for one-shot pruning, but one-shot pruning itself is problematic (will explain in a bit).
4) Table 1 is a poor representation, I would say, STR and GMP are extremely easy to implement. GMP could be done in One-shot and GMP is data-independent no matter what. I would like to hear authors' thoughts on this.
5) Coming to the setup, the biggest problem I have with the way GP or GPMT is being used is the application on top of a pre-trained network. Eg. On imagenet the fine-tuning of GP or GPMT takes 20 extra epochs, which isn't the case with all the baselines in the discussion. All of them are pruning while training baseline that only takes 100 epochs at best. I personally have adapted GP to be pruning while training like with IMP and GP does amazingly well, so the aspect of one-shot is overshadowed by the extra cost involved in fine-tuning. I don't know how we can make a case for these trade-offs to make the comparison fair.
6) The big one, the paper completely disregards the compute cost during inference (FLOPs) for these pruned models. The biggest difference GP and layer-wise pruning methods have is the inference costs involved. Please see STR or RigL papers to see about that axis of comparison. GP often has 2x more inference FLOPs than uniform-sparsity with very little gain in accuracy. If uniform sparsity was adjusted for that compute costs, the accuracy will be much higher. This paper, while being a study on GP, completely overlooks this aspect of GP or GPMT.
7) While the results on CIFAR-10 are good for debugging and understanding, the insights aren't particularly transferable. I would also not read too much into 40% sparsity results or even 95% sparsity because the networks involved are so heavily overparameterized that even at that sparsity level CIFAR-10 seems like a simple task. However, I would like to say Table 5 is a real thing for one-shot pruning, but will not happen for gradual GP. I would recommend authors implement gradual GP to see what it brings to the table, along with reduced training costs (debatable because one can think of pre-trained models as free).
8) Figures 2 and 3 just reinforce what I am saying above.
9) The gains in Tables 6 and 7 are not significant enough to make sweeping claims on SOTA while also not measuring FLOPs.
10) The use of "huge margin" is highly discouraged in Table 8 because the gains are very insignificant and trust me the FLOPs are going to be so exorbitantly high that the accuracy vs # params doesn't matter in that case.
11) As mentioned in the paper, MT doesn't seem to bring anything to the table for ImageNet experiments, just because of how the weight norms across layers change. MT is probably not a valuable addition in this case.
12) I would not get into the discussion of output preservation because, it doesn't necessarily mean that the model is doing well if its outputs are closer to some other model, the yardstick itself is flaky here.
13) Lastly, the discussion of MT tries to make it feel like it is not complex or data-dependent. But the same paper says GMP schedule is complex. I would say searching for MT is as good as coming up with a cubic pruning pattern which will even help GP when done in a gradual fashion.

Overall, I do not think the paper brings any new observations into the limelight and also ignores the issues present with whatever has been presented. The paper is not ready for publication, but I am happy to chat with the authors to help them understand and (myself) gain perspective if I am missing something.



**Summary Of The Paper:**

This paper revisits Global Pruning (GP) and makes a case to consider it seriously as part of the pruning literature. The authors also present an addition to GP, that claims, to further increase accuracy and reliability called Minimum Threshold (MT). MT is the constraint put on every layer of the neural network to ensure they are not pruned beyond a certain limit which might result in catastrophic accuracy drops.

The paper presents experiments on CIFAR-10 with a few CNN architectures along with current pruning baselines (layer-wise mostly. They also show results on ImageNet using ResNet50 and MovileNetV1. Following these experiments, the authors claim that GP or GPMT are SOTA for pruning despite their simplicity.

The paper also includes some discussion on how output patterns of each layer change with different pruning schemes and a note about how to set the optimal MT values.

While I agree with the sentiment of the paper, I do not think the paper is presenting anything new (even from a benchmarking perspective) but rather is missing the point about what makes GP vs layer-wise pruning a worthy trade-off often. I will list my concerns in the main review.

**Summary Of The Review:**

The paper does not propose anything novel (which is fine) but the observations are also not new while the paper ignores most of the issues and tries to underplay other baselines.

The paper is not ready to be published but is an excellent starting point for new readers.


=---------------------------------------------------------------------------------------

After an extremely long discussion with the authors and providing them with things I felt were necessary for fixing the experiments, claims etc., I think the paper is not ready to be published. The authors should revisit most aspects of the paper if they want to make it a benchmark paper for GP and claim so. None of the insights and contributions are novel given my discussion and experience. I  hope the authors understand that it is better to publish a ready paper than a paper changing around a lot at this point.

I vote for rejection.

---

> ### Author Response · Authors · 2021-11-16
> **Author Response (Part 3)**
>
> *7.	While the results on CIFAR-10 are good for debugging and understanding, the insights aren't particularly transferable. I would also not read too much into 40% sparsity results or even 95% sparsity because the networks involved are so heavily overparameterized that even at that sparsity level CIFAR-10 seems like a simple task. However, I would like to say Table 5 is a real thing for one-shot pruning, but will not happen for gradual GP. I would recommend authors implement gradual GP to see what it brings to the table, along with reduced training costs (debatable because one can think of pre-trained models as free).
> 8.	Figures 2 and 3 just reinforce what I am saying above.*
>
> We thank the reviewer for this suggestion. We are running the gradual GP for Table 5 and will report the results here once ready. We will then be able to evaluate if the problem of over-pruning is specific to one-shot.
>
> *9.	The gains in Tables 6 and 7 are not significant enough to make sweeping claims on SOTA while also not measuring FLOPs.
> 10.	The use of "huge margin" is highly discouraged in Table 8 because the gains are very insignificant and trust me the FLOPs are going to be so exorbitantly high that the accuracy vs # params doesn't matter in that case.*
>
> We would like to highlight here that # params and FLOPs are both important metrics that pruning tries to reduce. In most cases, there is a trade-off between # params and FLOPs (1, 2). Many methods explicitly have a goal of optimizing one of these metrics. Model size in itself is an important consideration for edge deployment and hence, should not be written off as a second tier optimization metric. For instance, for IoT devices, # params proves to be more important than FLOPs as it helps to limit expensive memory accesses. By keeping as many weights as possible in on-chip memory SRAM (usually on-chip memory capacity is very limited, especially for microcontrollers like STM series chips which only have less than 1MB or even 128KB memory), the memory access to off-chip DRAM will be minimized and thus significantly reduce latency/energy caused by the expensive off-chip memory access [3]. Thus, highly parameter sparse networks are very useful for many IoT applications running on microcontrollers embedded in battery-powered tiny device.
> We show that the GP method, does exceedingly well on # params on ImageNet. It achieves upto 1.4% higher accuracy on ImageNet vs. SOTA (Table 8, 95.3% sparsity), which we feel is very significant. Since, the main focus of our paper is to study the GP method in-depth, this is an important contribution that we feel the community should be aware of.
> 1.	What is the State of Neural Network Pruning?, Davis Blalock, Jose Javier Gonzalez Ortiz, Jonathan Frankle, John Guttag, MLSys, 2020
> 2.	MCUNet: Tiny Deep Learning on IoT Devices, Ji Lin, Wei-Ming Chen, Yujun Lin, John Cohn, Chuang Gan, Song Han, NeurIPS, 2020
> 3.	Efficient Processing of Deep Neural Networks: A Tutorial and Survey, Vivienne Sze, Yu-Hsin Chen, Tien-Ju Yang, Joel S. Emer, Proceedings of the IEEE, Volume: 105, Issue: 12, Dec. 2017
>
> *11.	As mentioned in the paper, MT doesn't seem to bring anything to the table for ImageNet experiments, just because of how the weight norms across layers change. MT is probably not a valuable addition in this case.*
>
> We would like to clarify that we never mentioned that MT doesn’t bring anything to the table for ImageNet. Quite contrarily, we show that MobileNet benefits greatly from MT on ImageNet at 90% sparsity. MT leads to an additional 4.5% accuracy gain, which is very significant for ImageNet (Table 9).
>
> *12.	I would not get into the discussion of output preservation because, it doesn't necessarily mean that the model is doing well if its outputs are closer to some other model, the yardstick itself is flaky here.*
>
> We adopt the above-mentioned yardstick from another paper - Lookahead: a far-sighted alternative of magnitude-based pruning, ICLR, 2020. It is indeed difficult to find a universal justification why a model is doing well, as there are different theoretical approaches to this. We adopted a one that is more relevant to our case i.e., for pruned models. We feel it is an interesting lens to evaluate and understand pruned models and is a good first step to provide another dimension along which to compare pruned models.
>
> *13.	Lastly, the discussion of MT tries to make it feel like it is not complex or data-dependent. But the same paper says GMP schedule is complex. I would say searching for MT is as good as coming up with a cubic pruning pattern which will even help GP when done in a gradual fashion.*
>
> We answer this in detail in point 4 earlier. Essentially GMP requires the search of 3 additional hyper-parameters compared to only 1 for MT. This is why we feel MT is simpler.

---

> > ### Comment · Reviewer_MHY2 · 2021-11-18
> > **Further continuation of the reviewer response**
> >
> > 7) (also 8) Let us talk after the results arrive. But my concern is that authors have not considered this to be a factor to experiment on when it is what they are comparing against in GMP. I would request authors to think deeply about the experimental setups before making sweeping claims (not that you make some here).
> >
> > 9) (also 10) Dear authors, I always said both are important, it is your paper that ignored FLOPs completely. Look at the 1st strength in my review, I mention for a given parameter budget to be the point in consideration. It is good that you bring real-world deployment into the picture. I have worked enough on actually deploying these models on tiny IoT devices with as little as 256KB Flash and 16 KB RAM. Yes, reducing #params is very important, but if I gain 1% accuracy (Which in real deployments might not even be perceivable with models of such small sizes, because of the relatively poor performance already) with a 2x increase in FLOPs, I will always opt for the faster model. Let us talk about ResNet50 models in your papers, none of the models except the ones at 98% are going to be deployable at the edge you are talking about but you are increasing 1% accuracy at a much more increase in FLOPs. It is a 3-way trade-off and all the 3 should be acknowledged. GP is SOTA for a given param count, that is something I noted no matter what and is not a novel contribution in this context. Let me know if you want to talk more about this.
> >
> > 11) I am sorry for the blanket statement on MT. I meant to say MT is not valuable on ResNets and then authors try to justify a potential cause. Let us talk about MT on MobileNets after we have gradual GP results.
> >
> > 12) I am not an expert on this and I feel that this analysis probably provides little information. I would defer to other reviewers for this part of the paper. I still don't think distances in activation space amount to anything for explanations or theoretical justifications.
> >
> > 13) Sure MT has only 1 hparam. But I have given you a set of hparams for GMP which will practically always work. That doesn't make MT simpler because you still search for it but now it is different for every architecture, while GMP's stuff probably generalizes even more.

---

> > > ### Author Response · Authors · 2021-11-21
> > > **Further continuation of author response**
> > >
> > > 7.Our experimental setups focus on comparing GP against SOTA methods. That’s the toughest set of baselines to compare against. Also, as evidenced by the new MobileNet experiment and what the reviewer themselves alluded to, gradual pruning improves the performance of GP even more. We believe these benefits can transfer to other architectures and datasets as well. Hence, gradual GP is likely to get even better results than one-shot GP. This only bolsters our empirical findings, that is, if one-shot GP outperforms SOTA on a constant parameter budget basis, then gradual GP will outperform even more. Hence, GP can be even more powerful and supports what we are trying to communicate in this paper.
> > >
> > > 8.We have acknowledged the lower performance on FLOPs in the updated version. We have added the FLOPs numbers to the table and have pointed it out in the text explanation, have added a limitations section discussing this, and even mentioned some mitigation strategy and future work to reduce FLOPs further. Kindly have a look.

---

> > > > ### Comment · Reviewer_MHY2 · 2021-11-21
> > > > **More Discussion Cont. Cont.**
> > > >
> > > > 7) Dear authors, you have claimed to compare against SOTA methods. I have provided you a simple baseline of gradual GP which outperforms all baselines including your own method. MobileNet experiment on CIFAR-10 does not count as a baseline to generalize insights, either way, so let us wait for ImageNet results. I never said GP is not powerful, I am saying the contributions and insights are not novel. The study is novel but is not at the scale of State of Sparsity etc., to become an anchor for future research.
> > > >
> > > > While it is good to evolve, now you see gradual GP is extremely strong and I am not sure about MT anymore. But MT was a key aspect of the pitch in the initial version. So, I am not fully sure about what the authors finally want to say. If it is that GP is strong and everyone should use it, I kinda agree (barring FLOPs) but I have been saying that when you are making a case for GP you should talk at lengths about limitations as well. see next point.
> > > >
> > > > 8) Thanks for adding FLOPs. I would like to point out that STR out pruned last layers even more than GP and still has much lesser FLOPs than GP. Just an observation.
> > > >
> > > > The fact that you are comparing FLOPs to DSR, SM, and RigL is not correct. They are sparse-to-sparse training methods and need way too many FLOPs during inference for various other reasons. I don't think this is the right statement to make.
> > > >
> > > > While talking about accuracy gains, did not talk about the near 2-3x increase in FLOPs for 1-2%. In the next version, I recommend the authors to drop the discussion about activation closeness and MT and actually talk about every aspect of GP and make it an actual study for empirical performance of GP, natural adaptations of GP, insights drawn from other papers based on their usage of GP etc.,
> > > >
> > > > Extending GP to gradual GP is not an exciting extension, it is a natural extension and it is bringing the same problem you are trying to answer that people ignore simple and important baselines.
> > > >
> > > > After all the discussion, I strongly ask the authors to go back to the drawing board, make a clear set of claims, and also fix the experimental setups to become a strong study for the future. At this stage, I am not able to push for acceptance.
> > > >
> > > > All the best and thanks.

---

> ### Author Response · Authors · 2021-11-16
> **Author Response (Part 2)**
>
> *4.	Table 1 is a poor representation, I would say, STR and GMP are extremely easy to implement. GMP could be done in One-shot and GMP is data-independent no matter what. I would like to hear authors' thoughts on this.*
>
> We mentioned in the Related Work section our rationale for giving the above ratings-
> 1.	STR incurs implementation complexity as it defines a custom convolutional module called STRConv (https://github.com/RAIVNLab/STR). It is not the vanilla Conv2d module as defined in PyTorch. The models generated by STR also cannot be directly opened in PyTorch. They require either modifying the existing codebases to be compatible with STR or convert STR models to standard PyTorch models. GMP requires a sparsification schedule to be set. The schedule comprises three important hyper-parameters that need to be set – the span of pruning steps (n), the starting training step (t0) and the pruning frequency (∆t) (https://arxiv.org/pdf/1710.01878.pdf). In contrast, GPMT does not require any modification to existing PyTorch functions or codebase and only requires 1 hyper-parameter to be set – MT itself. Hence, we find it simpler than the baselines. Having said that, GMP and STR are simpler to implement vs. AMC, which requires a RL training procedure to be implemented as well. The current format of our table only allows binary values (tick or not) and doesn’t allow us to capture the above range. Hence, we will update our table and reflect better the difference in ease of implementation of GMP and STR vs. AMC.
> 2.	Regarding GMP being one-shot, if it is made to be one-shot, then we would like to clarify if it would still be GMP? As it will not be gradual anymore?
> 3.	For data-dependence, the masks set by GMP in each epoch evolve based on changes in weights as found by Backprop. The current mask affects the forward pass, which affects the backward pass, which affects the subsequent mask that is calculated. Hence, Backprop plays a key role in the calculation of the final mask. Thus, we find GMP to be data-dependent. We hope this clarifies our thinking behind Table 1.
>
> *5.	Coming to the setup, the biggest problem I have with the way GP or GPMT is being used is the application on top of a pre-trained network. Eg. On imagenet the fine-tuning of GP or GPMT takes 20 extra epochs, which isn't the case with all the baselines in the discussion. All of them are pruning while training baseline that only takes 100 epochs at best. I personally have adapted GP to be pruning while training like with IMP and GP does amazingly well, so the aspect of one-shot is overshadowed by the extra cost involved in fine-tuning. I don't know how we can make a case for these trade-offs to make the comparison fair.*
>
> The choice of using 120 epochs for fine-tuning was inspired from previous works, specifically AMC (https://arxiv.org/abs/1802.03494). We actually did not try any other value for the fine-tuning epochs. In light of the reviewer’s comment regarding a fair comparison, we are re-running our ResNet-50 experiments on ImageNet using 100 epochs. We will report the results as soon as they are ready. We hope this will make the comparison fair.
>
> *6.	The big one, the paper completely disregards the compute cost during inference (FLOPs) for these pruned models. The biggest difference GP and layer-wise pruning methods have is the inference costs involved. Please see STR or RigL papers to see about that axis of comparison. GP often has 2x more inference FLOPs than uniform-sparsity with very little gain in accuracy. If uniform sparsity was adjusted for that compute costs, the accuracy will be much higher. This paper, while being a study on GP, completely overlooks this aspect of GP or GPMT.*
>
> We take note of the reviewer’s comment and have measured the FLOPs and find that GP does have higher FLOPs compared to the baselines. Our investigation also shows that the reason for this is that the layerwise algorithms prune the initial layers more (which have higher FLOPs) compared to GP which prunes the last layers more (which have less FLOPs). Hence, the overall number of FLOPs is lower for the layerwise algorithms. We will add an analysis section in the paper discussing the FLOPs behavior of GP along with the FLOPs results and upload the revised version before the rebuttal ends. Please also see response to points 9 and 10 for # params being an important goal for pruning.

---

> > ### Comment · Reviewer_MHY2 · 2021-11-18
> > **Continuation of the reviewer response**
> >
> > 4) I am very well aware and personally have used the methods mentioned in Table 1. The binary representation of properties is only valid if the properties are not subjective. Having things like simplicity is not factual but rather about perception. Let me give you the hparams for GMP which always work, start pruning at 20% of the epochs, end it at 80% of the epochs with a pruning frequency of 1 epoch. This will work for most of the cases and it really doesn't depend on the hparams after a point in the experimentation. AMC is in no comparison to what you are doing here, AMC brings so much else to the table by giving what you can control and what you want to reward, hence the complexity. STR does give some control but is kind of the Pareto optimal for FLOPs vs Model size vs accuracy. Not everything can be captured in the table by calling something simple. I reiterate Table 1 is not required in the paper to make the point you are trying to make.
> >
> > Sorry for the confusion about GMP. yes, when GMP is one-shot it is not gradual anymore, but that is same as one-shot layer-wise pruning which is not compared in this paper (CIFAR results are too few datapoints to compare on the small dataset anyway). Also GMP, like IMP can use any pruning strategy within itself, so GMP is not just about layer-wise pruning (it is an instantiation in that paper).
> >
> > GMP has 2 variants, one where the masks can change and the other where masks just contract over time. Both of them roughly work the same with a slight boost when allowed with changing mask patterns. Again, same choices can be made for Gradual GP when implemented. GMP's design choices and final sparsity patterns are not data-dependent either way.
> >
> > 5) AMC is a different story, and not exactly applicable right now, I hope the authors agree with it. Let us talk when the new results arrive.
> >
> > 6)  If there is a place you can use superlatives it would be the increase in FLOPs for the same accuracy across methods. Let us talk about this when you update the tables with FLOPs as promised. I will talk about params in #9 and #10, No one is denying #params being important.
> >
> > The investigation is already in STR, please look at the FLOPs and param distribution across layers for various methods which also includes a global method as a baseline in their paper. Not every layer-wise pruning algorithm prunes initial layers a lot and it is a trade-off, there is a sweet spot, GP is probably not it. The reason why GP prunes last layers a lot is because the "weight norms" of initial layers is higher and each weight has a higher magnitude. One way to think about is that all weights across all layers are equal hence magnitude is its importance, but intuitively it seems like not all layers are equal but rather params in a given layer are equal (Do read papers on "Are all layers created equal?", Random Pruning, ratios matter for sparse letter tickets for these arguments).

---

> > > ### Author Response · Authors · 2021-11-21
> > > **Continuation of author response**
> > >
> > > 4.(and 11) We have removed Table 1 as per the reviewer’s suggestion.
> > >
> > > 6.We have studied the characteristics of different layers as well. We find that the average weights across the first 4-5 layers and last 2-3 layers for instance, on a ResNet-50 are much higher than other layers and remain like that post pruning as well. There is not much correlation between the size of the layers and the weight magnitudes as the last layers are usually quite big but still have higher magnitude weights. This may also be the reason why many pruning approaches don’t or only partially prune the first and last layers. This does indicate the implicit importance of weight magnitudes as a pruning criteria. We also thank the reviewer for sharing the paper links.

---

> > > > ### Comment · Reviewer_MHY2 · 2021-11-21
> > > > **More discussion cont.**
> > > >
> > > > 4) Thanks for removing Table 1.
> > > >
> > > > 6) Dear authors, Again, there seems to be some miscommunication. If the weight magnitudes of last layers are good enough to not get pruned, then what is the point of MT? I point you to Fig 6 of STR paper, where GS is the sparsity budget they obtained from global pruning. The last layers are not as heavily pruned as layers right before them. I never said anything about size of the layers and weight norms, but there indeed is a correlation that not all layers have weights of equal norms, so it might not be a great idea to use weight magnitudes as proxy for pruning as there might be a tiny bottleneck layer with extreme small weight norm that gets pruned out. I hope this makes sense?

---

> ### Author Response · Authors · 2021-11-16
> **Author Response (Part 1)**
>
> We would like to thank reviewer MHY2 for reviewing our paper and for their very detailed comments. We are also thankful to them for noting the key strengths of our paper which is to show that GP while being such a simple pruning algorithm is also able to achieve SOTA pruning performance. Below, we provide explanations to the points raised by the reviewer.
>
> *1.	No one ever claimed GP is not sufficient to achieve SOTA performance on pruning. GP is extremely powerful no matter what gives the highest accuracy (often) for the same #params even at the highest sparsity levels. GP is not forgotten or overlooked by any means. The reason why people work on layer-wise pruning is slightly different and I will come to it later.*
>
> It’s heartening to hear the reviewer’s thoughts on GP, although we feel that in the larger pruning community GP is either used as a baseline or not even known at all. While its clear that the reviewer is very familiar with GP, we feel that this is not the case for the larger research community. We already detail in the Related Work section how very few research papers mention GP at all and those that do, mention it only as a baseline. Furthermore, we are not familiar with any research papers that showcase that GP can achieve SOTA pruning performance or give the highest accuracy vs. SOTA layerwise methods for the same number of params. To our knowledge, this is the first work to do a systematic analysis of GP and actually showcase that it can achieve SOTA pruning performance vs. layerwise SOTA methods. We feel that these results and insights will be valuable to researchers and the wider pruning community.
>
> *2.	While the authors mentioned people of LTH community probably used GP as one of the potential pruning techniques, GP was the only one that resulted in strong tickets at higher sparsities, making IMP explicitly work on GP no matter what other factors are.*
>
> As the reviewer noted themself, LTH only utilized GP at higher sparsities. It did not use GP for all their experiments. Even then, LTH was not a study on GP. They did not compare GP with SOTA layerwise techniques to evaluate how good or bad GP is compared to layerwise techniques. In contrast, the focus of our work is to spotlight GP. We perform detailed experiments on GP and answer the question of how GP compares to layerwise SOTA pruning methods.
>
> *3.	MT comes into the picture only when there is one-shot pruning and GP has been shown to work great even with gradual schedules (IMP is an example) and will not suffer from pruning out entire layers just because of the gradual processing. I would even argue setting the small pruning schedule is data-independent and has a similar cost as finding the optimal MT as done in this paper. MT makes sense for one-shot pruning, but one-shot pruning itself is problematic (will explain in a bit).*
>
> We have not tried using a gradual schedule with GP and we find the reviewer’s suggestion that over-pruning will not happen in a gradual schedule very interesting. We will try to implement the gradual schedule and report back the results here with regards to over-pruning.

---

> > ### Comment · Reviewer_MHY2 · 2021-11-18
> > **Thanks for the rebuttal**
> >
> > This is a very thorough rebuttal. I will respond to each part separately. I might sound critical at times but I would ask the authors to read in-between the lines and try to get the gist whenever possible.
> >
> > We both agree on the same things but value the contributions differently. Let me respond back to each of your rebuttals.
> >
> > 1) Yes, the community might be ignoring GP as a baseline, but the community also ignores reporting FLOPs or actual inference time which matters in deployment. I am trying to find papers that show global pruning is extremely strong, this might be one of the latest papers which shows that: https://arxiv.org/abs/2106.09857 and uses GP for non-uniform sparsity. So goes with the choice of GP as the basis of IMP for LTH among many other things. Yes, the insights are valuable (I didn't deny that), I am arguing that insights are not new and might not count as a contribution. Yes, your systematic analysis is correct, but that doesn't mean it is a contribution as things are not new, even if they are buried and being ignored, they still are not new.
> >
> > 2) I did not understand LTH not using GP for all their experiments (some clarification would help). LTH used layer-wise and GP as the underlying algos in IMP at lower sparsities and saw that performance didn't change, however, the accuracy changed for higher sparsities and they needed to use GP to make it work. This also happens to work only because the FLOPs of GP is so much more during inference, sort of hinting at higher expressivity. LTH is not the work to talk about the value of GP explicitly. But they implicitly made design choices showing the power of GP. GP always is an intrinsic choice.
> >
> > Regarding how GP compares to layer-wise pruning. There is no comparison between one-shot layer-wise and one-shot GP (there are 2 datapoints on cifar-10 and that can't be enough anyway, even in that case, I think GP is going to be better than layer-wise (uniform)). GP shines super well when combined with gradual pruning (as IMP shows). GP will have higher accuracy than layer-wise (unless there is a budget design for layers happening around). GP will also have higher FLOPs (up to 2x) compared to layer-wise methods. GP is probably not the pareto optimal when all factors are combined. These are insights I would expect from GP no matter what.
> >
> > 3) Let us talk about MT stuff after the new results arrive.

---

> > > ### Author Response · Authors · 2021-11-21
> > > **Revised manuscript and response to the reviewer**
> > >
> > > We would like to thank reviewer MHY2 for their in-depth reply and appreciate their time in replying back to us. We have modified the paper as per the reviewer’s suggestion. The key changes made are-
> > > 1.	Removed Table 1
> > > 2.	Added Gradual GP result to Table 4
> > > 3.	Updated numbers for 100 epochs instead of 120 in Table 7
> > > 4.	Added Flops to Table 7 and Table 8
> > > 5.	Added limitation and future work section
> > >
> > > We have also highlighted the changes in the manuscript in yellow for easier reference of the reviewers. For the other comments, kindly see our answers below.
> > >
> > > 1.We would firstly like to emphasize that the insights presented by us are indeed new. The suggested paper is about grow-and-prune methods - “Effective Model Sparsification by Scheduled Grow-and-Prune Methods”. They do not present results from GP alone and hence, we cannot conclude anything about GP’s performance from their results. Our key contribution is to show that GP can indeed outperform SOTA algorithms on accuracy. No other paper shows this and hence, the insights are indeed new.
> > >
> > > 2.(a) The following paragraph from LTH demonstrates their view towards GP very succinctly – “Global pruning. On Lenet and Conv-2/4/6, we prune each layer separately at the same rate. For Resnet-18 and VGG-19, we modify this strategy slightly: we prune these deeper networks globally, removing the lowest-magnitude weights collectively across all convolutional layers. In Appendix I.1, we find that global pruning identifies smaller winning tickets for Resnet-18 and VGG-19. Our conjectured explanation for this behavior is as follows: For these deeper networks, some layers have far more parameters than others. For example, the first two convolutional layers of VGG-19 have 1728 and 36864 parameters, while the last has 2.35 million. When all layers are pruned at the same rate, these smaller layers become bottlenecks, preventing us from identifying the smallest possible winning tickets. Global pruning makes it possible to avoid this pitfall.” As is clear, they switched to GP because they ran into the problem of over-pruning with uniform sparsity. The only insight that can be drawn from their approach is that GP is better than uniform sparsity at higher sparsities. No conclusion can be drawn from this on whether GP is more powerful than other SOTA pruning algorithms. And thereby this does not showcase the power of GP. As opposed to this we clearly showcase that GP outperforms SOTA algorithms across a range of architectures, datasets and sparsities, and thereby show that GP is indeed powerful.
> > > (b) We would like to point the reviewer to Tables 1, 2, 5 and 6. Uniform sparsity and SNIP are both one-shot methods.

---

> > > > ### Comment · Reviewer_MHY2 · 2021-11-21
> > > > **More discussion**
> > > >
> > > > Thanks for the response. Here are my thoughts and I still would urge the authors to read in between the lines at times, not everything could be literal in this case and some of the statements you attribute to me might not be what I said. As earlier, I might sound supercritical at places but I request the authors to read things carefully. The fact that something like MT was rendered moot with a simple suggestion I made should help in having some assurance in my comments.
> > > >
> > > > Thanks for removing Table 1. Updating epochs to 100. Adding FLOPs and limitations. While it is good to see gradual GP in Table 4, that is not where you claimed the most about MT, so I would wait till the results of gradual GP for ImageNet turn up to make any claim about MT whatsoever.
> > > >
> > > > 1) I know “Effective Model Sparsification by Scheduled Grow-and-Prune Methods” does not present insights for GP. It uses GP for non-uniform sparsity and shows extremely good results. GP is a design choice that made things work. I reiterate GP outperforming SOTA algorithms on accuracy is not a new contribution, not having a paper show does not really mean insights are indeed new. How do you explain the fact that I am saying all the things GP could do if it were not for acquired insights from the papers proposing IMP etc., While GP has not been surveyed liked GMP was in Gale et al., 2019 the insights of GP are implied by a lot of design choices across papers no matter what. You can indeed gather insights from such papers which don't just focus on GP.
> > > >
> > > > 2) Uniform sparsity is a SOTA algorithm depending on where you use it. GP and uniform sparsity fall into the same design choice buckets. I would argue IMP did in fact showcase the power of GP. SNIP is a different method altogether it is more worried about sparse-to-sparse training and that is another direction that we do not want to open now as the baselines need to be much more carefully thought out.

---

> ### Author Response · Authors · 2021-11-24
> **Thanks for the discussion**
>
> We thank the reviewer for their comments and insights. While we have differing opinions with the reviewer on some aspects of the work (as discussed in the rebuttal), we nevertheless take the feedback seriously and appreciate the suggestions offered by the reviewer. Thank you.

---

### Official Review · Reviewer_5P4J · 2021-11-03

**Correctness:** 3
**Technical Novelty And Significance:** 3
**Empirical Novelty And Significance:** 2
**Recommendation:** 6
**Confidence:** 2

**Main Review:**

* Layer-wise sparsity ratios used to be tricky hyper-parameters. A global threshold and a minimum number of preserved weights eliminate layer-wise sparsity ratios. In this context, the proposed method is practical.

* Although the empirical results seem promising despite of the simplicity. it is not straightforward to understand why we should use global threshold across layers and why MT is crucial to ensure superior performance for certain models. For example, how does over-pruning impact the performance?

* A related study [1] argues that layer-wise sparsity is of importance. A comparison (both theoretical and empirical) should make the result stronger.


[1] Lee, Jaeho, et al. "Layer-adaptive Sparsity for the Magnitude-based Pruning." International Conference on Learning Representations. 2020.

**Summary Of The Paper:**

A very simple and effective pruning method is proposed. Instead of layer-wise pruning, a global threshold is used to prune weights according to their magnitude. In addition, to avoid over-pruning, a minimum number of parameters is preserved for each layer after pruning. Experiments on CIFAR-10 and ImageNet validate the effectiveness of proposed method.

**Summary Of The Review:**

The simplicity and effectiveness of proposed method are important and interesting. The findings will inspire more studies on the nature of network pruning. However, it is not easy for readers to understand where the effectiveness of proposed method comes.

---

> ### Author Response · Authors · 2021-11-16
> **Author Response**
>
> We would like to thank reviewer 5P4J for reviewing our paper and for noting the key merits of this approach. Below, we provide answers to the questions raised by the reviewer.
>
> *1.	Although the empirical results seem promising despite of the simplicity. it is not straightforward to understand why we should use global threshold across layers and why MT is crucial to ensure superior performance for certain models. For example, how does over-pruning impact the performance?*
>
> The simplest case to understand why over-pruning is problematic is the case when a whole layer is pruned away. In such a scenario, the network essentially gets split into two components that are not connected to each other. Hence, no information flows from one component to another, and the network does not learn anything and gives out random chance accuracy (Table 5, where accuracy for GP on MobileNet-V2 is 10% only). Hence, pruning a whole layer crashes the network. Similarly, over-pruning to such an extent that only a few weights (typically <1000) are left in a layer creates a big bottleneck for information to flow through the network and may not be sufficient for feature information from a previous layer to flow to the next layer. The Minimum Description Length (MDL) may be a good framework to understand this concept better, where a certain minimum number of bits is required to represent the information fully.
>
> *2.	A related study [1] argues that layer-wise sparsity is of importance. A comparison (both theoretical and empirical) should make the result stronger.*
>
> We looked at study [1] but were unable to find where a case for layer-wise over global pruning is made. Could the reviewer point us to the exact argument they are referring to? Nevertheless, we added a section in our paper (Section 5.2) to give a more theoretical comparison between layerwise pruning and GP. We compare the per layer Frobenius output distortion of the layerwise and GP models, as proposed by another study (Equation 2, [2]). We measure which of the two models has lower distortion in outputs compared to the original model. We find that GP is able to preserve outputs much closer to the original model as compared to the layerwise model (Fig. 5). As can be seen, the layerwise model has much higher distortion than GP especially in layers 1 and 2. We plan to add even more theoretical insights into the differences between GP and layer-wise pruning and is in our plan for future work.
>
> [2] Lookahead: a far-sighted alternative of magnitude-based pruning. Sejun Park, Jaeho Lee, Sangwoo Mo, and Jinwoo Shin., International Conference on Learning Representations, 2020.

---

### Official Review · Reviewer_YX1q · 2021-11-03

**Correctness:** 3
**Technical Novelty And Significance:** 2
**Empirical Novelty And Significance:** Not applicable
**Recommendation:** 5
**Confidence:** 5

**Main Review:**

1) My major concern is that the paper does not clearly explain differences between GP in this paper and traditional GP methods. Based on Equation (1), I think GP in this paper is the same as GP adopted in Han et al.(2015a). If GP in this paper can achieve SOTA, Han's method can also achieve SOTA. My suggestion is that the authors provide qualitative or quantitative comparison with previous GP methods.

2) The proposed MT is not a sufficient improvement. First, GP+MT increases accuracy on MobileNet, while slightly decreasing accuracy on ResNet. Second, I am not sure whether MT is sensitive to selected threshold.

3) Since CIFAR-10 is a tiny-scale dataset, I think experiments on CIFAR-10 cannot sufficiently validate that for WideResNet-22-8, GP+MT can further improve accuracy compared to GP with the same sparsity ratio.

**Summary Of The Paper:**

The paper revisits a traditional model compression method, global magnitude pruning (GP), and shows that GP can achieves state-of-the-art. The paper further improves GP by introducing minimum threshold (MT). Experiments on ResNet show that GP+MT achieves better accuracy with the same sparsity ratio compared to GP.



**Summary Of The Review:**

The paper does not clearly explain differences between GP in this paper and traditional GP methods. The proposed MT is not a sufficient improvement. So my rating is "5: marginally below the acceptance threshold"

---

> ### Author Response · Authors · 2021-11-16
> **Author Response**
>
> We would like to thank reviewer YX1q for reviewing our paper and their comments. We would like to clarify that we show that in cases where GP over-prunes a layer, MT can easily fix the over-pruning issue (e.g., MobileNet on ImageNet). MT is not a flat addition that needs to be applied to every model, for instance, on ResNet-50 on ImageNet, there is no need to apply MT and GP alone works very well. We would also like to clarify that the main focus of our paper is GP and not MT. In particular, we show that such a simple algorithm like GP, which is often ignored, beats all the SOTA pruning algorithms on a range of architectures and datasets, including ImageNet. This gives pruning researchers new insight on what is required for a good pruning scheme (i.e., weight magnitudes ranked at the global level) and sets a strong baseline for future pruning works to follow.
> Below, we provide detailed explanations to the 3 points raised by the reviewer.
>
> *1.	My major concern is that the paper does not clearly explain differences between GP in this paper and traditional GP methods. Based on Equation (1), I think GP in this paper is the same as GP adopted in Han et al.(2015a). If GP in this paper can achieve SOTA, Han's method can also achieve SOTA. My suggestion is that the authors provide qualitative or quantitative comparison with previous GP methods.*
>
> Our GP method (excluding the addition MT) is exactly the same as traditional GP methods i.e., to rank weights in the whole network by their magnitude and then prune the bottom x%. We explain this in Section 3.1 in the paper. We would also like to clarify that Han et al. (2015a) do not use GP. They instead use layer-wise pruning. See section 5 in their paper for a detailed discussion on how they choose a layer-wise threshold. An excerpt from their paper – “We used the sensitivity results to find each layer’s threshold: for example, the smallest threshold was applied to the most sensitive layer, which is the first convolutional layer.” We hope that clarifies the reviewer’s doubt.
>
>
> *2.	The proposed MT is not a sufficient improvement. First, GP+MT increases accuracy on MobileNet, while slightly decreasing accuracy on ResNet. Second, I am not sure whether MT is sensitive to selected threshold.*
>
> We provide detailed discussion on how MT works and why it is only required to be added in certain architectures like MobileNet (Section 5.1 and 5.3). Essentially MT prevents a layer from being completely pruned away or pruned to a very high extent. Thus, its especially useful for very small starting models (e.g., MobileNet) or very high sparsity rates. Hence, there is no need to add MT to every pruning scenario and it is a good thing that MT does not need to be added to every pruning scenario as GP alone is sufficient in many scenarios. Secondly, MT is sensitive to threshold, and we address that in the paper. We have provided MT values for all our experiments in the first table in Appendix section A.4. Also, it is to be noted that for our algorithm MT is the only additional hyperparameter to be tuned, in contrast to tuning multiple additional hyperparameters in other existing methods e.g., tuning the span of pruning steps (n), the starting training step (t0) and the pruning frequency (∆t) for GMP.
>
>
> *3.	Since CIFAR-10 is a tiny-scale dataset, I think experiments on CIFAR-10 cannot sufficiently validate that for WideResNet-22-8, GP+MT can further improve accuracy compared to GP with the same sparsity ratio.*
>
> MT improves performance on WideResNet by 0.21% at 95% sparsity and 0.74% at 99.9% sparsity. Also, in case the reviewer is comparing the WideResNet gains vs MobileNet, then it is to be noted that gains on MobileNet are bigger because a whole layer is pruned away for MobileNet and the accuracy crashes. MT helps to recover this. But in WideResNet, the network is never disconnected and accuracy doesn’t crash, hence, there is no accuracy to recover. This is explained in detail in section 5.1.
> Lastly, as highlighted previously, the focus of this paper is not to add MT to every pruning scenario but to show that for certain cases (where a layer is over-pruned), MT can be added to GP to boost its performance. Other examples for this is MobileNet-V1 on ImageNet, where adding MT improves performance by upto 4.5%.

---

### Decision · Program_Chairs · 2022-01-20

**Decision:**

Reject

**Comment:**

The paper claims that one of the most common (and obvious) pruning methods in the literature today (global magnitude pruning) is "overlooked" and "seen as a mediocre baseline by the community." As an active member of the pruning research community myself, I can attest that this is simply not true. I am in strong agreement with reviewer MHY2 and - after reading the discussion around that review and the paper itself in detail - I confidently recommend rejection.

Magnitude pruning itself dates back decades, at least to the work of Janowski (Pruning vs. Clipping in Neural Networks, 1988). The paper is correct that *global* magnitude pruning (in which all weights are compared in a layer-agnostic manner) was largely ignored in favor of layer-wise magnitude pruning (i.e., pruning all layers by the same amount) in much of the work that popularized magnitude pruning (e.g., Han et al., 2015). However, global magnitude pruning has become much more popular since that time. In work establishing the lottery ticket hypothesis, Frankle and Carbin (The Lottery Ticket Hypothesis) use it in certain cases and - later - in all cases (Frankle et al., Linear Mode Connectivity and the Lottery Ticket Hypothesis). In the past several years, global pruning in general has become the de facto way to use all new pruning heuristics (e.g., SNIP: Single-Shot Network Pruning based on Connection Sensitivity; Picking Winning Tickets Before Training by Preserving Gradient Flow; Pruning Neural Networks without Any Data by Iteratively Conserving Synaptic Flow). Moreover, other papers have specifically advocated that global magnitude pruning is state-of-the-art within recent years at this very conference: Comparing Rewinding and Fine-Tuning in Neural Network Pruning (Renda et al., ICLR 2020 oral): "We propose a pruning algorithm...that matches state-of-the-art tradeoffs between Accuracy and Parameter-Efficiency across networks and datasets:...globally prune the 20% of weights with the lowest magnitudes." (This paper does not cite Renda et al. despite the fact that it is a prominent paper that directly contradicts the purported problem that the paper relies on to support the significance of the findings.)

In short, in the pruning literature, the idea that global pruning, magnitude pruning, or global magnitude pruning is overlooked or is not recognized as a strong baseline is simply preposterous. The reason that global magnitude pruning has "largely been ignored in recent years, generally being relegated to the position of a baseline for comparison" is because it is a simple technique whose efficacy has long been known and established - exactly what a good "baseline for comparison" should be.

The paper has narrowed its claims somewhat during the discussion and revision period, advocating for a one-shot global magnitude pruning strategy that "does not require any complex pruning frameworks like RL or sparsification schedules [or]...iterative procedure." To do so, however, the proposed method replaces each of these "complex" hyperparameters with another set: whether or not to use a minimum threshold (MT) and where to set it. Even if the approach isn't iterative, the hyperparameter search necessary to set it almost certainly is, and it is unclear whether searching for the MT value is any more efficient than the other approaches. The costs of this hyperparameter search need to be measured. And iterative pruning's costs can often be mitigated by making pruning gradual, something the paper considers superficially in the revisions.

Finally, as reviewer MHY2 observes, one of the primary reason papers *don't* use global magnitude pruning is that, although it leads to higher sparsities than layerwise magnitude pruning, it also often leads to higher FLOP counts. Although FLOP counts are a terrible indicator of real-world speedup, they are a much higher-fidelity indicator than parameter-count, which neglects the fact that - in convolutional networks - a small number of parameters can lead to vastly more FLOPs if they operate on larger activation maps (i.e., before the activation maps have been downsampled). In the revisions, the paper gives a token nod (and a superficial dismissal) to this fact in Sections 4.3 and 6, but the paper needs to fully acknowledge this point by measuring and discussing its consequences. "Look[ing] at this in future work" is not enough.

Due to these many concerns, I strongly recommend rejection.